# Performance Evaluation of a Lane Correction Module Stress Test: A Field Test of Tesla Model 3

Jonathan Lancelot , Bhaskar P. Rimal * and Edward M. Dennis

The Beacom College of Computer and Cyber Sciences, Dakota State University, Madison, SD 57042, USA;
jonathan.lancelot@trojans.dsu.edu (J.L.); edward.dennis@dsu.edu (E.M.D.)
* Correspondence: bhaskar.rimal@dsu.edu

**Abstract:** This paper is designed to explicate and analyze data acquired from experimental field tests of a Tesla Model 3 lane correction module within the vehicle's Autopilot Suite, a component of Tesla OS. The initial problem was discovered during a nominal drive of the Tesla Model 3, where after a random number of lane correction events, the lane correction module shuts down, issues a visual disable warning on the touchscreen, and control of the vehicle is given to the driver until the next drive. That development was considered problematic, as the driver can be caught off guard or may be medically disabled and unable to respond. During a controlled stress test, a more severe issue was discovered. After a random number of lane correction events, the lane correction module shuts down without warning, then stays activated after the test driver corrects the vehicle's trajectory. This is considered a fatal error in the system and adds a dangerous element to an otherwise standard feature in a modern automotive vehicle. The results established that the number of events needed to trigger a fatal error without warning is unpredictable. Our results also demonstrate that the system is inconsistent.

**Keywords:** cybersecurity; physical security; human security; human safety; electric vehicles; Internet of Things

## 1. Introduction

Historically, the patent for a camera-based lane correction system was filed on 12 August 1999 by K. Schofield [1]. This invention relates to object detection adjacent to a motor vehicle as it travels along a highway and, more particularly, relates to imaging systems that view the blind spot adjacent to a vehicle and/or view the lane adjacent to the side of a vehicle and/or view the lane behind or in front of the vehicle as it travels down a highway. Tesla moved away from the radar and made the engineering decision to build vehicles, starting in 2021, relying on Tesla Vision, their camera-based autopilot system [1]. Any Tesla manufactured before 2021 had radar-integrated systems, and the project did not have access to these earlier vehicles. This project had access to a Tesla Model 3 Performance 2022, equipped with Tesla Vision only, and a Tesla Model 3 Long Range 2021, provided with Tesla Vision only. The former was the primary stress-testing vehicle, and the latter was for external verification that the fatal error was potentially an issue across the fleet.

Specifically, the focus of the stress test was on the timing out of the module after a certain number of instances of lane correction events. First, lane assist provides steering interventions if Model 3 drifts into (or close to) an adjacent lane where an object, such as a vehicle, is detected. In these situations, Model 3 automatically steers to a safer position in the driving lane. Second, lane departure avoidance is designed to warn you if Model 3 drifts out of or near the driving lane's edge [2]. Both program components are combined into the module that is being tested in the field. The goal was to understand how many lane correction events occur before the module times out and disables itself without warning, how random the numbers of lane corrections events per test are, and if the timing out of

the module poses a danger if the operator is incapacitated, endangering other operators on the road. The module cannot detect the reason for the lane correction having multiple consecutive events. Therefore, if the operator is having an issue with consciousness or a medical emergency, and the system disables or has a fatal error, the probability of an accident is high if the vehicle requires manual control. Even if the operator of the vehicle is healthy, if the module times out randomly, the probability of a vehicular accident involving the vehicle or multiple vehicles is likely, because the driver becomes psychologically conditioned by the module to drive with lane correction over time, and a sudden change in driving characteristic can disorient even the most experienced drivers. This last point is beyond the scope of this paper.

The study has noted that Tesla issued a disclaimer on the feature by stating that users should never depend on lane assist to inform if the operator has unintentionally driven outside of the driving lane or to inform that there is a vehicle beside or in a blind spot [2]. The limitations and inaccuracies are acknowledged as there is an awareness that the system continues developing as it is being used in a real-world applications. Tesla issued other warnings that future studies will critique, yet the larger concern is that inaccuracies are life-threatening if the application is used in the first place. Lane correction is not an application that is typically used unless the driver is in need of the module activation, which is the prerogative of the operator, and expects the module to function until the operator deactivates it.

There are also issues with notifications to the driver when the driver needs to take over from the autonomous driving software. The National Highway Transportation Safety Administration (NHTSA) has been researching distracted and drowsy driving. Suppose the driver is not paying close attention, whether distracted or drowsy while driving. In that case, they may not catch that the autonomous feature is relinquishing control of the vehicle back to the driver. In 2016, fatal traffic crash data [3] showed 3450 fatalities due to distracted driving and 803 drowsy-driving deaths. It is difficult for law enforcement to identify drowsy or distracted driving instances in the event of an accident. The number of deaths due to distracted or drowsy driving is challenging to quantify, and the NHTSA continues to work on quantifying it. However, it is still a concern if the driver is to acknowledge and react to the autonomous feature releasing control. The driver not seeing that they need to take control could prove fatal. A distracted or drowsy driver may miss the alert on the management screen.

The implication of this study stems from the discovery of a fatal error during stress testing, and Artificial Intelligence (AI) and Machine Learning (ML) will play a significant role in transportation, specifically in the automotive industry. The context here is human security and safety from the perspective of an operator and the assumptions made by developers on the intent and medical condition of the operator of the vehicle. The implications of testing automation elements in driving are critical to developing smart AI that recognizes when an operator has an emergency and can clarify and act on the emergency by evading potential accidents. This is if the system warns the operator explicitly on the touchscreen inside the vehicle. There is no warning in the case of a fatal error discovered during the stress test. Currently, Tesla Full Self-Driving (FSD) is a feature that is under strict scrutiny because of the number of car accidents involving Tesla vehicles and the autopilot software/hardware usage by the operators. The U.S. Department of Justice launched the previously undisclosed probe last year following more than a dozen crashes, some of them fatal, involving Tesla's driver assistance system Autopilot, which was activated during the accidents [4]. Aside from the company being investigated, the study did not set out to stress test the whole Autopilot Suite as the study is taking one component of the AI capability, lane correction, and testing it against manual driving, which the module is designed to correct. The FSD feature is not the focus of this stress test, as the error in the lane correction module in the Autopilot Suite was discovered during preliminary testing. Tesla is not the only car manufacturer that has dealt with failure issues in lane correction technology features. Note that a class-action lawsuit in 2021 alleged that the systems that make up

Subaru's EyeSight Driver Assist Technology suite of safety features in the 2013–2021 vehicle models are hampered by software calibration and integration issues and do not work as advertised in real-world driving conditions. The 187-page lawsuit alleged Subaru failed to warn buyers and lessees that the vehicles' pre-collision and reverse automatic braking features, which together make up the automaker's autonomous emergency braking system (AEB), and "Lane Keep Assist" functions, posed a real safety risk [5]. The issue that affected the "Lane Keep Assist" module in the Subaru model was a system-wide issue that affected all components of the EyeSight Driver Assist Systems components, designed for interoperability. Our stress test on Tesla's lane correction module was outside the scope of research to determine if the fatal error discovered was linked to issues with FSD and the total Autopilot Suite.

Although the automotive industry and public traffic administrations are planning for automatized road traffic, its introduction will ultimately depend on how public attitudes develop [6]. As an experimental field research operation, it is significant that the lab had a Tesla Model 3 to perform the action research method for direct data acquisition. The operation had one Model 3 for primary testing and another for external verification of the error in the lane correction module. The outcome sought through field test research is to develop a solution or recommendation for Tesla to develop an emergency feature in case of no response from an operator instead of a time out that seems arbitrary and random. However, during the stress test, an error was discovered, requiring an update or repair of the module source code first. The significance of this research is that human security and safety solutions for the manufacturer will be deliverable. Towards this end, the contributions of this work are summarized as follows:

- A method of stress testing and collection of data in experimental test driving was developed that allowed for the discovery of the fatal error.
- Our field test exposed an error in the lane correction module that will provide results to confirm that the lane correction module needs further testing from the manufacturer.
- This study merged human security and safety by testing the cyber–physical relationship between the operator and the vehicle, including resultant vehicle behavior.
- This study will create awareness of the importance of rigorous testing of vehicle software and hardware post production.

The remaining segments of the paper are structured as follows: Section 2 briefly discusses related work. Section 3 discusses the testing methodology of the stress test. Section 4 presents the data, their classification, and the analysis. Finally, Section 6 concludes the paper and provides some recommendations.

## 2. Related Work

In [7], authors investigated how to influence elements in a vehicle's proximity until they trigger a collision in simulation. Their focus was on validation by simulation with a particular emphasis on building realistic models of the environment, including models of driving behavior and sensors. However, reference [7] does not cover the element of human factors, how human interaction and the unique designs of human software developers influence the cyber–physical relationship between the vehicle operator and the vehicle, and the vehicle and the multi-variable stochastic nature of real-world road testing. It is agreed that road tests alone cannot give a full view of all of the unknown variables and possibilities for malfunction in real-time. We realize that every situation, vehicle manufacturer's vehicular product including software, human driver, and every study in this field are unique, which makes this research particularly challenging. This is why our stress test was focused on giving a narrow recommendation for a specific manufacturer who developed a specific lane correction module that is being used in real-world situations.

In [8], authors investigated the naivety of simulations where the space of driving scenarios is too expansive for sampling techniques to cover adequately, and real-world vehicle testing is commonly employed for autonomous vehicle validation. Still, the costs and time requirements are high. Nevertheless, the missing element is the actual cost to

back the proposition that the cost of real-world testing, including time requirements, is prohibitive. In [9], the authors investigated whether simulations can offer an inexpensive complement to field testing for evaluating the safety of autonomous vehicles.

In [10], authors investigated how we could prove that an autonomous vehicle can drive in live traffic. Their goal was to expose the disastrous consequences of improper testing. In parallel, we discovered in our stress test that, indeed, improper testing could potentially lead to accidents in the real world if manufacturers do not develop methods of stress testing that are designed to expose dangerous inconsistencies. Nonetheless, studies that attempt to research whole autonomous systems instead of studies that employ a method of component or module isolation run the risk of missing critical errors in the simplest portions of intelligent software. In [11], authors examined a novel method for generating test scenarios for a black box autonomous system demonstrating critical transition in its performance modes. In [12], the authors inspected the question of if we can deploy a fleet of fully autonomous driving systems that are safe enough to leave humans completely out of the driving loop. However, the important question is what is impeding us from taking humans out of the loop of driving just yet? The study conducted by our institution can inform researchers that the auto industry is not prepared to move in the direction of a driverless car without placing individuals who use the automotive transportation system at considerable risk. Today, we must contend with human error on the road and human error in software development. What is being proposed is taking human error off the roads without adequate development of a testing system for human error in the software development process. Our work is different as we are dealing with practical testing of real systems which can produce fatal errors in the real world.

In [13], authors studied the problem of testing advanced driver-assistance systems (ADAS) and the promise of a reduction in the number and severity of traffic accidents, traffic congestion, and fuel consumption, thus leading to resource saving. Notwithstanding, this study is taking on three very complex components of the problem they are trying to solve. A study that investigates each of these components using their metric would yield a clearer focus for researchers in the industry attempting to develop technology to improve the safety of ADAS. Our work is focused on a single component of a larger complex system, which allowed for an exposure of a fatal error that might not have been discovered if we were stress testing the Autopilot Suite on a whole system.

In [14], the authors proposed a method that is designed to evaluate lane departure correction systems effectively. This paper functions as a guide for our initial test of the hypothesis and allows for the development of an evaluation model. The paper states that these systems can potentially address many serious injuries and fatal crashes. Yet, if the lane correction module malfunctions under a stress test, there is a concern that a module designed to lower instances of serious injuries and fatal crashes could increase them. The authors in [15] proposed an improved lane determination algorithm to increase the accuracy and stability of the driving assistance system in an automatic driving system when determining the position of a lane in a complex driving environment [15]. This paper provides a general understanding of how algorithms can influence events and determine when lane assistance is disabled or an error altogether. The work in [16] provides a proposal for a lane-keeping assistance system, which is also called a lane departure prevention system. Our work differs as our research is a real-world field test on a specific vehicle for a specific manufacturer, where we can give practical results and recommendations on a lane correction module sold to consumers. This guides our research by illustrating the theoretical architecture of a lane correction module since there was no access to investigate Tesla's software architecture.

## 3. Test and Methods of Stressing Lane Correction Module

This study addresses a research gap by attempting to expose lane correction in terms of the lack of human security and safety built into the software after the feature is disabled. Yet, during the stress test, a fatal error in the lane correction module of the Tesla

Model 3 was discovered during the preliminary stress test by lead researcher and experimental test driver Jonathan Lancelot, who conducted the field test in a real-world environment. If the vehicle operator is not responsive and lane correction detection is required, a consecutive set of events in a nominal instance should not time out. If the feature is disabled with a warning, it requires a manual vehicle takeover. However, what happened during the stress test was lane correction module errors, comparable to a buffer overflow, and without warning, it switched to manual control and gradually rotated towards the off-road. The probability of a traffic accident is high. This hypothesis is proven true, and the solution and recommendation will require a patch or update from the manufacturer or some other remedy if the issue is a software or hardware issue. The details of the experimental arrangement are to first get the vehicle to the appropriate speed and center between the lanes. Second, to release the steering wheel, keep your foot on the accelerator (spike), and observe the vehicle drift to the left or the right lane. Third, observe the vehicle bouncing off the line, and count the instances. Fourth, be prepared for the module to fail and for the vehicle to move over the lane off the side of the road. If the module fails, swiftly grab the wheel and center the vehicle between the lanes to repeat the experiment. Lastly, activate autopilot (autopilot will keep the vehicle centered while the test driver records results) to record the data while looking at the road, then at the computer for half-a-second intervals until data are recorded before deactivating autopilot and continuing with the experiment. Other research teams can reproduce this arrangement to develop their results and compare.

### 3.1. Testing Design Framework

Every testing instance has an input, output, controls, resources external to the experiment, and the component of test configuration, speed requirements, test identification, and collected data. The test drivers' input triggers output from the vehicle, and the tests are designed to filter the real-world variables that are extraneous and in need of environmental control variables.

Figure 1 shows a schematic of testing and data collection. There are two sources of vehicle testing methodologies that guided the approach of the field test. Rigorous testing of vehicular computers is critical to security and safety. Therefore, functional tests, security, and safety protocols for the familiarization phase of a component stress test, and these segments of their method were adopted from [17]. At a set point in an operational procedure, when the familiarization phase of testing is complete, the analyst should have an in-depth understanding of (i) hardware operation and interfaces, (ii) separation of software and hardware processes, and (iii) safety-critical requirements. First, familiarization with hardware operation and interfaces are required for developing stress-testing methods and mastering the dynamics of vehicle performance and capabilities. Second, the disaggregation of software and hardware processes is an important analytical step for test drivers to estimate where the failure starts and when it ends during a stress test. Lastly, the critical safety requirements for field testing include, for example, safety gear for the test driver and the rules of engagement in real-world conditions. The stress test was designed to induce consecutive lane correction events to seek uniformity in the deactivation procedure, as it was known that a deactivation event is inevitable.

In the black-box approach to testing, i.e., distributed development, configuration, and maintenance in the automotive supply chain, one has no or only very limited access to knowledge about concrete and internal system implementation, and a lack of data about system architecture, configuration files, and the source code of the system under test (SUT) [18]. The reality of manufacturers' concerns with intellectual property (IP) can hinder deep software research that does not require driving the vehicle; however, a cyber–physical test that requires an experimental test drive to stress certain critical components, especially components that are computer-driven, is a practical form of cybersecurity testing that focuses on physical security and human safety.

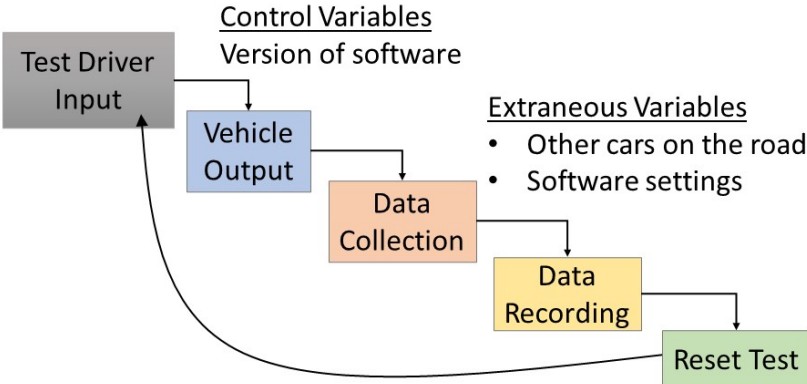

**Figure 1.** Testing design framework.

An important element of the testing framework is shown in Figure 1, where the design supports observing and collecting instances of lane correction error events. The development of data classification is also important for active research, as it creates a knowledge basis for the continuous testing of the lane correction module beyond the scope of this paper. The test driver would notice the disabling events if the module were active, explicit, or implicit. Tacit knowledge was gained by the test drive by the continuous use of the vehicle before and during preliminary testing, which set the basis for measured observation and hypothesis development. As the research design and methodology allow for the reproduction of the error in question, upon collecting data from each test, the development of a solution and recommendation for the vehicle manufacturer is required.

The testing framework was designed to formulate a method of isolating the problem, enumerate occurrences of the problem, and analyze the data to determine if the lane correction module needs to be revisited with a fix. In the method of test driving the vehicle, for instance, the test driver acts as a monitor while the car (set to manual) drifts towards the right or left sideline of the road without intervention. The human factor in this test is designed to replicate the scenario where there is no reaction from the human operator, and the vehicle becomes an immediate danger to the life of motorists within proximity. During a test, a functioning lane correction instance does not require intervention from the test driver, yet the occurrence of a fatal error or a malfunctioning lane correction instance does. A method was developed to record the functional and malfunctioning instances during the test. The test driver input these instances into a `csv` file on a temporary and independent computer system that was brought onboard. The testing framework reflects the design of the physical stress test, which informs the development of the data classification and data recording methods in creating this framework, which is a quantitative test designed to isolate the problem and observe the rate of occurrence. Figure 2 is a sample photo of footage from live recordings of the stress tests from the test driver's perspective. It shows Jonathan Lancelot entering the result of the fatal error post-occurrence as the vehicle is set to autopilot to allow for the timely entry of data. The test driver consistently monitored the road for safety at one-second intervals.

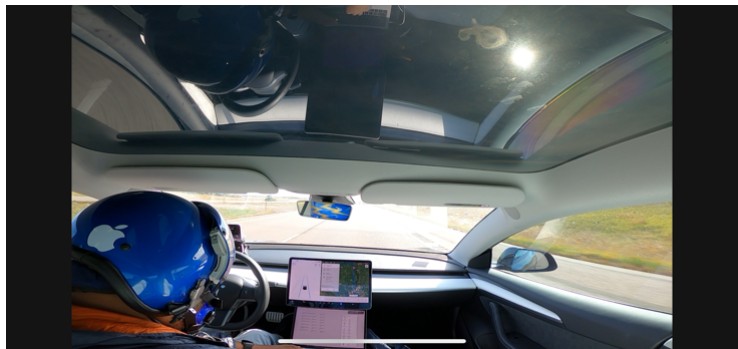

**Figure 2.** Footage from the preliminary field test.

*3.2. Test Driver's Notes on Tests and Methods*

There were a few instances where the vehicle had to be placed into autopilot to record the correction instances and the instance of an error in the `csv` file created for data collection. The basic autopilot feature, ironically, was stable enough as a trusted application to record the malfunction of the lane correction module. Next, it was discovered early in the preliminary testing phase of research that the vehicle had issues bouncing from the right side of the road to the left or vice versa. Another anomaly was encountered in the same phase, which will take further tests to confirm the occasional malfunction of the lane correction module when the right sideline disappeared due to intersections in the road. The testing framework overall is a predictive model that calculates the behavior of the vehicular system regarding lane correction and attempts to anticipate the random cadence of the error event. Nevertheless, what was discovered was continued randomness, as there were no distinctive patterns of errors discovered overall.

The field test experiment was chosen to focus on the relative reliability of a segment of the Autopilot Suite (lane correction module) designed to make the vehicle's operation safer, not increase the risk, despite the legal disclaimer. Figure 1 illustrates a workflow of the field test. The workflow consists of (i) selecting the test permutation, (ii) positioning the vehicle at the designated speed and center within the lane, (iii) with all lane correction settings in place and the vehicle set to manual control, releasing the steering wheel, (iv) the test driver observing the lane correction events, (v) capturing the steering wheel after the fatal error output, (vi) recording results and (vii) resetting for the next test. The workflow mirrors the testing design framework, as the test driver is an executor of the framework as the vehicle needs to be stressed to output results.

## 4. Data Collection and Classification

The experiment was conducted as a field test, in which the lead author was the test driver and data collector. The subject vehicle type is a Tesla Model 3 Performance 2022 as the main test vehicle, and a Tesla Model 3 Long Range 2021 was used for external validation of the problem. Both are running software version 2022.20.18. There were 43 successful tests conducted between 6, 7, and 10 October 2022. The testing framework was designed to expose the random behavior of a fatal error occurrence. The problem needs to be precisely formulated and justified by showing that it is significant and potentially dangerous for real-world applications [19]. The initial problem was the inconsistent behavior of the lane correction system disabling itself after a random number of lane correction events on the road, which was accompanied by a warning on the touchscreen (see Figure 3). The preliminary test was conducted, and another dependent variable was discovered, where a fatal error occurred without warning the touchscreen. The secondary vehicle was tested as a means of external validation of the fatal error discovered with the main test vehicle.

The new hypothesis was a consistency problem in the lane correction feature in Tesla's software, which disabled itself without warning after a random number of lane corrections.

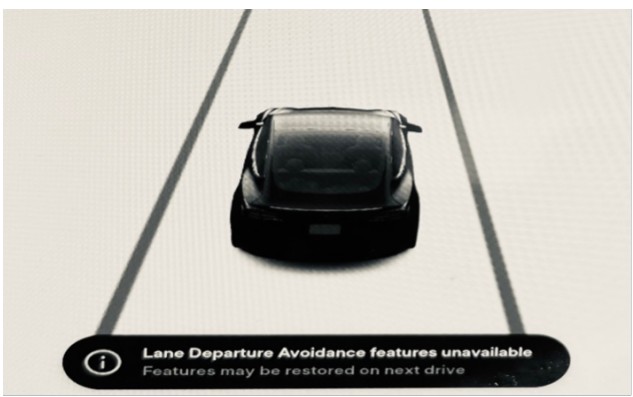

**Figure 3.** Disable warning output on a touchscreen by Tesla's computer after a random number of correction events.

The independent variable is the number of lane correction events triggered before the lane correction is disabled without warning and the feature is no longer activated. The dependent variable is the delivery of the error and deactivation of lane correction. To measure this relationship, three separate lane correction test permutations were created. The different test-driving permutations were designed to capture error-trigger events, for example,

- $P_1$: Lane correction active, car set to manual, foot on accelerator (spike), hands off the steering wheel.
- $P_2$: Lane correction active, car set to manual, foot off the spike, activate cruise control, hands off the steering wheel.
- $P_3$: Lane correction active, car set to manual, foot off the spike, no cruise control, hands off the steering wheel. The car should come to a complete stop.

Therefore, each test can contain one to three permutations, depending on scope, time, and budget. Given this field experiment's scope, time, and budget, $P_1$ was the only permutation used. Additionally, each test will require environmental controls in place, as this is a field experiment in a real-world situation. These permutation test categories are needed to allow for the expansion of research to account for different types of driving situations.

*4.1. Control and Extraneous Variables*

The version of software installed in the vehicle is the sole control variable to consider for this experiment, and all the extraneous variables include (i) other cars on the road, (ii) software features and settings, (iii) weather, and (iv) vehicle speed. First, the control variable has been articulated to be the version of software installed in the vehicle during testing. Even though we set these limitations, it is plausible to estimate that the observed problem is persistent and involves hardware and software. The question of external validity was verified and approximated with the second test vehicle, Tesla Model 3 Long Range 2021, using $P_1$. Updating the software of the vehicles is an important step in eliminating the possibility of a software issue alone, and this was verified on both test vehicles. In other words, the issue could have involved hardware in a complex computing system, where other variables come into play, yet this was unseen during field experiments. These variables were not considered to be confounding or intervening at the cyber–physical/driver–vehicle level, as no data indicating otherwise were discovered.

Second, extraneous variables were discovered during preliminary testing that disrupted data acquisition or delayed the acquisition within a scope of a limited time frame to complete the test. Vehicle field tests can be affected by other cars on the road, as the test was scrubbed if a car came too close from behind our test vehicle or a car moved in the opposite direction next to the test vehicle on a two-way road. When a test was scrubbed because

of proximity issues, the test was repeated, and the session continued. Other extraneous variables are software features in the autopilot section on the UI console.

As seen in Figure 4, all modules were activated. This setting was looked at as a possible factor affecting the outcome of the experiment. It was confirmed that developing different permutations of the I/O settings could confirm or determine if the change was null or negligible for this stress test. The stress test used the same configuration during the test as there were time constraints. The lane correction avoidance module must be set to assist and not warn or turn off at all times during testing.

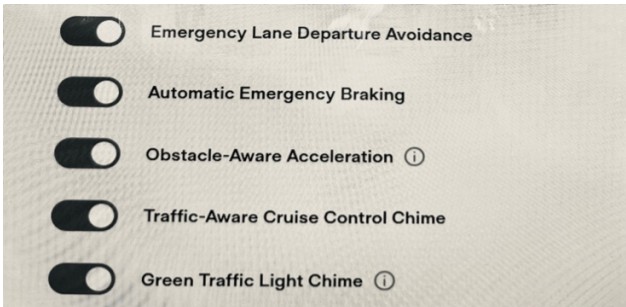

**Figure 4.** Autopilot section of Tesla UI.

The third extraneous variable is the condition of the weather. We limited our testing to dry weather with no precipitation, as the physical nature of the test could change and create a dangerous situation for real-world testing. In addition to this determination, testing in weather conditions should be a specific test that focuses on lane correction consistency in precipitous weather and its effect on the correction matrix, which should be pursued with the design presented here or with an improved version of the experiment. The test will be scrubbed if heavy rain or severe weather surface on test day.

The fourth extraneous variable could be the speed of the vehicle. This could be mitigated by measuring and limiting the speed level of each test, which would add three to six tests. The three speeds will be approximately 25, 27, and 29 meters per second (m/s). The controlling variable of road rules and laws constrained our ability to test at speeds lower than 25 m/s and higher than 29 m/s, yet this is the natural environment where real-world events occur. The interval will be denoted by $x \geq 25 \leq 27 \leq 29$, where the velocity can be set below 25 m/s yet cannot be above 29 m/s due to traffic law constraints. This limits the search scope for instances that reproduce the error in lane correction behavior. Instances of error will be found within the velocity limits given. The test was carried out on live highways in the state of South Dakota in the U.S., and away from traffic, as there was enough road to conduct the test without any traffic in the area for miles.

The test data classification system model was created for the stress test (see Table 1). The numeral system used for classification is, for example, test data classification being represented by number 2.0.1, where the number 2.0 represents the speed requirement, and 1 represents the test configuration. In this example, the speed is 25 ms/s, and test configuration $P_1$. Four tests (6.0.1, 7.0.1, 8.0.1, and 9.0.1) were selected for this test because of budget and time constraints. The first column enumeration system marks the test and its test configuration and is designed for use by the experimental test driver and future experimental test drivers for data collection. The speed requirement in the second column is for data collection and classification purposes, and the third column is for connecting the test results to the configuration. Lastly, when the vehicle is in the middle of a test, the hazard lights switch on, and the test driver has to switch them off to not startle others in traffic when they are in proximity. In a controlled road environment, the test driver would keep them on, and it is unclear if the vehicle does anything else after it switches on the hazards when there is no response. This variable is unknown until further tests are conducted or the vehicle manufacturer confirms what happens if the hazards are kept on.

**Table 1.** Test data classification system for field test data acquisition.

| Test Data Classification | Speed Requirement (ms/s) | Test Configuration |
|---|---|---|
| 1.0.1 | 25 | P1 |
| 2.0.1 | 25 | P1 |
| 3.0.1 | 25 | P1 |
| 4.0.1 | 27 | P1 |
| 5.0.1 | 27 | P1 |
| 6.0.1 | 27 | P1 |
| 7.0.1 | 29 | P1 |
| 8.0.1 | 29 | P1 |
| 9.0.1 | 29 | P1 |
| 1.0.2 | 25 | P2 |
| 2.0.2 | 25 | P2 |
| 3.0.2 | 25 | P2 |
| 4.0.2 | 27 | P2 |
| 5.0.2 | 27 | P2 |
| 6.0.2 | 27 | P2 |
| 7.0.2 | 29 | P2 |
| 8.0.2 | 29 | P2 |
| 9.0.2 | 29 | P2 |
| 1.0.3 | 25 | P3 |
| 2.0.3 | 25 | P3 |
| 3.0.3 | 25 | P3 |
| 4.0.3 | 27 | P3 |
| 5.0.3 | 27 | P3 |
| 6.0.3 | 27 | P3 |
| 7.0.3 | 29 | P3 |
| 8.0.3 | 29 | P3 |
| 9.0.3 | 29 | P3 |

*4.2. Preliminary Testing*

During the initial round of preliminary field testing, we conducted a $P_1$ test and sought to trigger a lane correction disable event with manually induced lane correction events.

Table 2 shows sample data collected from a test drive. The first test had five lane correction instances, one feature error and no disabled events; the second test had ten instances with one feature error; and the third with three instances with one feature error. The feature error was discovered during preliminary testing and changed the research focus. The feature disable on the lane correction application is when it alerts the driver that the feature has been disabled. The feature error is when the application disables itself with no indication, warning, or alert. The latter is the most severe, and the scope of this research is focused on the error. The disabling feature has a different solution requirement than the feature error, and during testing, it was found that the feature being disabled where the driver is warned was rare during stress tests.

**Table 2.** Data collection spreadsheet for test drivers.

| Lane Correction Instances | Feature Error | Feature Disable |
|:---:|:---:|:---:|
| 5 | −1 | 0 |
| 10 | −1 | 0 |
| 3 | −1 | 0 |

The initial problem under examination is how Tesla's autopilot, FSD, and lane correction modules go directly to manual control if they disable themselves automatically. The logic is that if the driver is not paying attention or is distracted, the human driver is forced to take complete control of the vehicle. The problem with this software is that the vehicle's AI assumes that the human is healthy and not distracted because of sickness or loss of consciousness. However, it being disabled without warning could catch any driver off guard.

With the discovery of the module error, it can be concluded that when the lane correction module is stressed, it produces a fatal error. To confirm this, we conducted a stress test on a secondary test vehicle (Tesla Model 3 Long Range 2021) to confirm that this issue is not limited to the main test vehicle, allowing external validity to be acquired.

In Figure 5, the schematic demonstrates how the Autopilot Suite, lane correction module, and FSD module are disabled, and the vehicle goes into manual mode. All roads lead to manual control if autopilot or any of its modules is disabled. Nevertheless, it is noticeable that lane correction is the only component with an error path. The reason for this is that lane correction was the focus of the experiment, and an error was discovered during specific stress testing of the module in a real-world environment. Disables and errors lead back to the manual vehicle takeover by the human driver, which is the central problem being examined. Yet, the main concern is the unpredictability of the feature error, leading to manual control without a warning or checking the driver's condition. The recommended solution model will be built on a standard software fix or recall.

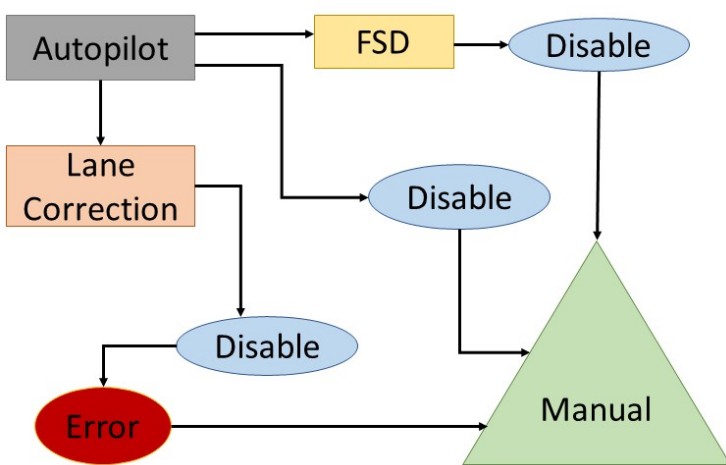

**Figure 5.** Schematic for Tesla Autopilot Suite, test driver's perspective.

There are limitations that were recognized at the beginning of the field test, one being that even if we set parameters to control variables within the real-world environment, it was found that it was difficult to stay at a specific velocity in a traffic environment and performing a multi-configuration test using permutations for $P_1$, $P_2$, and $P_3$ would be better suited for a controlled road experiment or independent race track to collect data without fear of interfering or disrupting the flow of real traffic. These results were all completed in a $P_1$ configuration, and the error in lane correction was reproducible enough due to the data to show the issue would be consistent in that mode. Regardless of the limitation, it is reasonable to deduce that this issue is not limited to one vehicle; however, there is an awareness that

a secondary $P_1$ test would be on a secondary vehicle of the same model. Yet, the trim is not as crucial except if the study had the allocated time and access to the software needed. Despite these limitations, the results ended up discovering a fatal error in the lane correction module, as it often occurred at random intervals when stressed.

Once the data are collected, statistical methods can be used to gain some predictability in terms of the average number of lane correction events during a singular test and other ways of predicting the occurrence of an error during a stress test. It should be noted that there is anecdotal evidence of drivers experiencing a disabling of the module during a nominal driving session. What is meant by anecdotal is an a posteriori experience of the event without the deliberate collection of data or intent to collect data for research purposes.

There is a possibility of using an application for reliability analysis, yet it could be a superficial model if we need access to the car's back-end application chatter. Therefore, we can use simple statistical methods, such as sample variance since the fatal error is clear and not ambiguous:

$$s^2 = 1/(n-1)(\sum_{i=1}^{n} X_i^2 - n\bar{A}^2) \tag{1}$$

and sample standard deviation equation,

$$s = \sqrt{1/(n-1)(\sum_{i=1}^{n} X_i^2 - n\bar{A}^2)} \tag{2}$$

in an attempt to develop a different angle on the exposure of the problem. A sample means formula will also be used to obtain a general view of the problem [7]. The tools used to analyze the results exposed the depth of the failure of the lane correction module. Table 3 is a representation of a metric designed to measure the accuracy of a lane correction module, allowing for the grading of the module's performance during stress testing.

**Table 3.** Accuracy rating percentage.

| Catagory Description | Accuracy Rate Range |
|:---:|:---:|
| Pass | (95–100)% |
| Moderate | (90–94)% |
| Fail | 89% and below |

## 5. Experimental Results

For the analysis of random occurrences of fatal errors discovered during the initial field test, the Notion of a Random Variable statistical analysis model was best suited to capture the randomness of events. The model verifies that the test design capturing errors is quantitatively precise. A random variable X is a function that assigns a real number, $X(\mu)$, to each outcome $\mu$ in the sample space of a random experiment. Recall that a function is simply a rule for assigning a numerical value to each set element, as shown below in Figure 6. The measurement specification on the outcome of a random experiment defines a function on the sample space and hence a random variable [x]. Each test starting consecutively with classification number 6.0.1 and ending with 9.0.1 is considered a sample space denoted by $S_x = 6.0.1, 7.0.1, 8.0.1,$ and $9.0.1$.

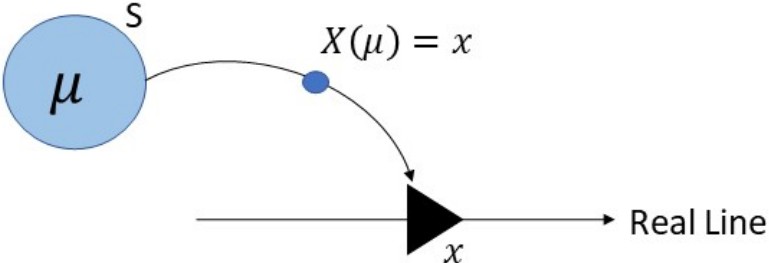

**Figure 6.** A random variable assigns a number $X(\mu)$ to each outcome $\mu$ in the sample space S of a random experiment [x].

In Figure 6, $\chi(\mu) = \mu$ is the relationship between the number of lane corrections and the error. For example, to check that the data collection method was quantitatively repeatable, $\chi(\mu)$ is denoted by the number of lane corrections, and $\mu$ for the module error. Say we want to demonstrate that data were collected where they can be verified within a statistical framework.

$$S = \chi(\mu) = \mu \tag{3}$$

When randomness is observable, it can be measured against what is expected as a normal operation, in this case, a machine or function. Using a standard deviation equation will give more metadata behind the statistical nature of the numerical spread and the confidence interval.

Now we have established a statistical notion of the random variable, which is the error discovered during the test, we can assume $\chi(\mu) = T_l c$ and $\mu = T_e r$. It must be noted that errors in a set of lane corrections are denoted by -1. For the calculations of the error rate and accuracy rate of the lane correction module, we use $|x|$, which is the absolute value of any negative integer calculations. This is the formula that will be used to process results in field tests 6.0.1, 7.0.1, 8.0.1, and 9.0.1. This is illustrated in Figures 7–10. The total sample size of data collected from the experiment cumulatively is 359 lane correction events, with 43 of them ending in a fatal error event. Therefore, approximately 12% of events cumulatively are errors. As a result, approximately 88% of lane correction events are non-errors.

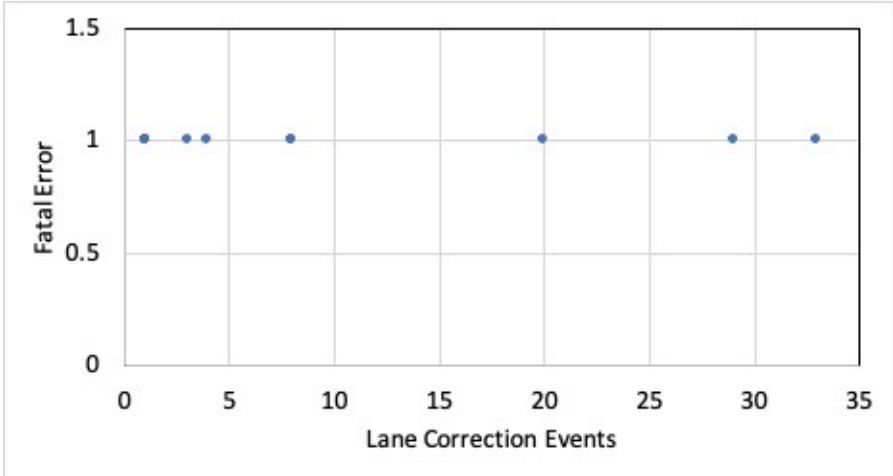

**Figure 7.** Fatal error vs. lane corrections events for test 6.0.1.

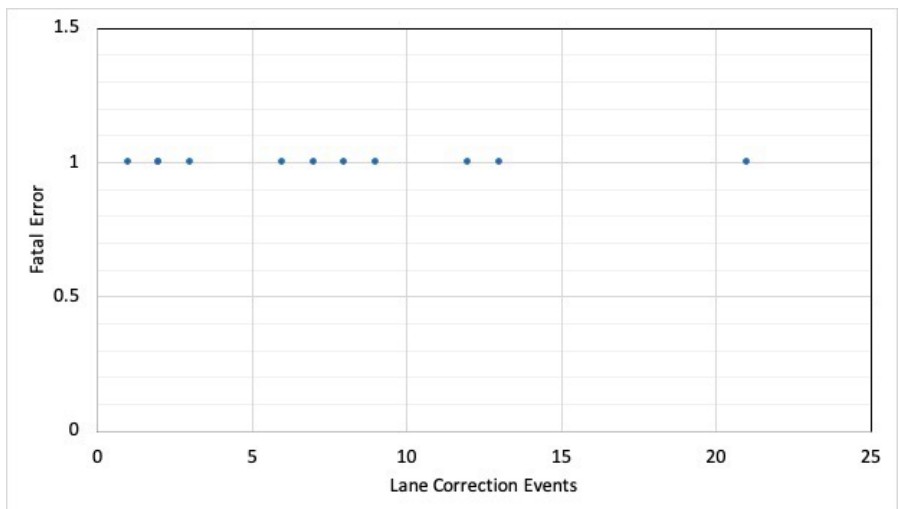

**Figure 8.** Fatal error vs. lane corrections events for test 7.0.1.

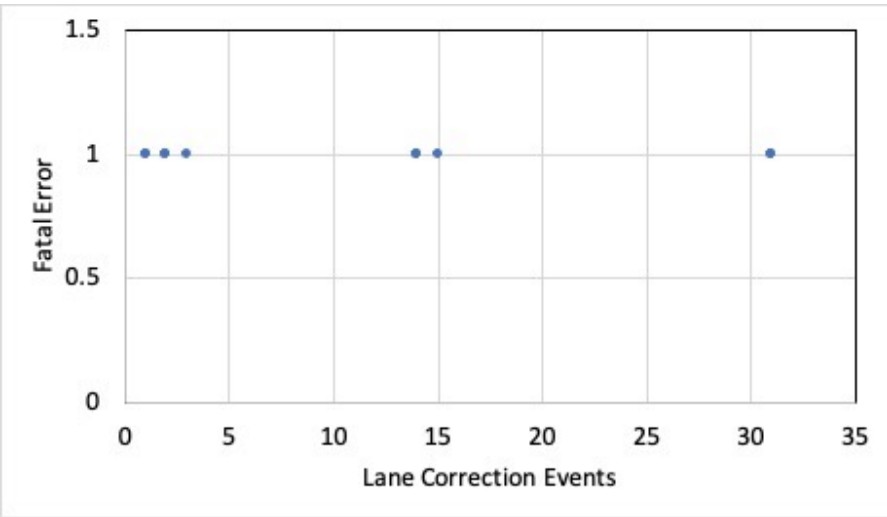

**Figure 9.** Fatal error vs. lane corrections events for test 8.0.1.

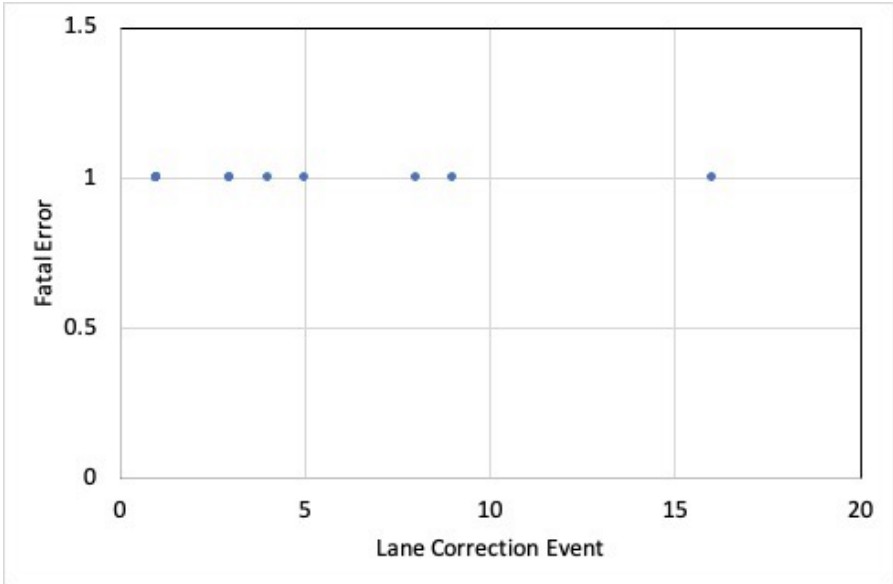

**Figure 10.** Fatal error vs. lane corrections events for test 9.0.1.

In test $S_{6.0.1}$ (see Figure 7), we can calculate the error rate overall by adding all total lane corrections, so $T_{lc} = 108$ and $T_{er} = 10$, which gives us an error rate ratio of 10.8 and an error rate percentage of 9.26 in all. If we were to isolate one event within this testing instance, we would find that the error rate varies significantly, giving another view of the random nature of the error events. For example, if we take $T_{lc} = 3$ and $T_{er} = 1$, we obtain an error rate percentage of 33.3, yet if we isolate from the test $T_{lc} = 29$ and $T_{er} = 1$, we receive an error rate percentage of 3.45.

Next, we want to establish the accuracy rate of the lane correction module in test 6.0.1; when we add all events within the test, we obtain a rating of 90 percent, according to Table 3. This is barely a moderate accuracy rating and 1 percentage point away from a failing grade. It can be assumed that more events would have yielded a failure as they are consistent across lane correction events. If we calculate the accuracy rate on each instance separately, we will find there are more failures per instance than a pass. Interestingly, the standard deviation of this data set is 12.1 and it has a confidence interval of 3.8, which can also represent a module failure.

Some anomalies were noted that are worth mentioning. First, on the instance where $T_{lc} = 33$ and $T_{er} = 1$, for the first time in the test, the disable function activated with a proper warning signal in the user interface screen. This was the initial hypothesis that was being investigated. Yet, a more severe error was discovered that was more consistent, a deactivation of the lane correction without any warning at all. It was also noticed that there were three instances where the error occurred on the first run, which indicated complete deactivation without indication. The vehicle's speed was approximately 29 m/s, yet it does not seem to have a noticeable effect on the erratic behavior of the lane correction module.

In $S_{7.0.1}$ (see Figure 8), the same methods will be used to determine the error rate. If we take the sum of all error events in the test, we have $T_{lc} = 84$ and $T_{er} = 11$, which gives us a ratio of 7.65, which is an error rate of 13.1%. Again, if we work with the error rate formula with each error event individually, we will see the erratic behavior of the lane correction module in percentages. Next, we look at the accuracy rate of the lane correction module. When we plug in the values of all events and errors combined, we receive an accuracy rate of 87%, which is a module failure according to the metrics employed. When we break down the events and calculate them individually, the reason for a higher percentage in 7.0.1 than in 6.0.1 will be apparent. The standard deviation of this data set is 6, and it has a confidence interval of 1.8. As a note on this particular testing session, the car allowed us to see how the car performs during a bounce from the right-hand side of the road to the left. In this instance, the lane correction module error occurs after the first bounce off the right lane of the road and fails on the left. When the steering wheel of the vehicle is released and the car is drifting, centrifugal force favors the right-side solid lane due to the slight convex contour of the road. When the car drifts, bounces, and rotates to the left-hand side of the perforated lane line, the module error occurs. This is a real limitation, as the condition and contour of the road can alter the physics of vehicular trajectory. Yet, the data are valid as the vehicle should not switch to manual during a two-sided bounce, which can send the car into oncoming traffic, endangering drivers.

Test $S_{8.0.1}$ (see Figure 9) has an error rate of 8.8% cumulatively, and when broken down into individual error events, the random nature of the error occurrence can be seen. When the test was broken down into individual error events, we confirmed a set of four repeat percentages, from which we can speculate that the randomness of the error events has been reduced. When the accuracy rate for the lane correction module is calculated cumulatively, the percentage is 91.2, a moderate rating, not a pass or fail. When we calculate the accuracy rate of individual error events, we verify that most events receive a failing grade. The standard deviation and confidence interval are 11.8 and 3.7, respectively. It was noticed that the car's visual system encountered a break in the right solid line, and sometimes there were errors, and in other instances, there were no errors, and the cars bounced off the line nominally. In other words, errors occur even where there is no break in the line. It was recorded that during an error event, a broken line on the road was

encountered by the visual system, yet there are not enough data to find a correlation. Subsequently, on one of the error occurrences, the car did a two-sided bounce. It crossed over the left perforated line, a critical failure that, if the driver is incapacitated, could cause a head-on collision.

Test $S_{9.0.3}$ (see Figure 10) is where the lane correction test failure was the most severe. Cumulatively, the error rate ratio yielded for the test is 4.41, and the error rate is 23%. Individual calculations for each error event yielded 5 instances of 100 % failure of the lane correction module out of 12 error instances. In other words, there were five instances where failure was encountered on the first try. The accuracy rate of the lane correction module is 77.4% cumulatively, which is a failure. When the error events are broken down individually, 11 out of 12 events fail. The standard deviation is 4.6, and the confidence interval is 1.32.

Notes from the field test confirmed that there were multiple events where an error occurred on the first instance of the car touching the right solid lane line. It was also noted that testing the module on the left-hand side of the road (American), in which a perforated line is present on a two-lane road, was not tested at the same frequency as the right-hand side of the lane was tested because of safety reasons. It proved too dangerous to test without a controlled road environment. However, it seems like the lane correction module fails to detect the left sideline during a two-sided bounce from right to left. It was successful only once. In one of the error events, it was observed that there was a crossroad intersection (where the sideline disappeared) that was passed during the test. It is unclear if that contributed to the error, as errors occured in a variety of road positions and conditions. There were two instances on the test where the vehicle bounced from the right to the left lane and failed to correct itself on the left lane side. The speed varied around the limit set by the experiment requirements, yet the car's speed did not seem to impact the outcome in prior observations during initial practice runs.

The conclusion for the failure of the lane correction module is based on the numerical analysis demonstrating the randomness of the error event and from the perspective of a driver depending on consistent operational behavior from a vehicular system. The strict scientific method applied to the experiment gives an observable structure for engineers attempting to solve the problem and allows for an objective view of the technical issue without the burden of brand loyalty or commercial popularity. There is a comparison test that the Human and Autonomy Laboratory conducted at Duke University in 2020, where they conducted field tests on multiple system modules for the reliability of driver-assist functions. The lane correction module was one of the features of the ADAS system that was tested. The conclusion of their test, which was cumulatively compared to our singular and practical focus on the lane correction module, is that the performance of the computer vision systems was extremely variable. This variation was likely responsible for some, but not all, of the delays in alerting a driver whose hands were not on the steering wheel [20]. Their conclusion on Model 3, focusing on the software and hardware performance of the computer vision system version at that time, aligns with the results and conclusion found in this experiment. Yet, our method of stress testing lane correction was designed exclusively for the component instead of a cumulative test on multiple systems.

## 6. Conclusions and Outlook

It can be deduced that the issue discovered by this stress test can be linked to industry software development practices. Additionally, field testing was the best option for the cyber–physical system approach to this experiment because these vehicles are designed for human drivers, who can have medical complications that might require lane correction to be as consistent as possible until either FSD is activated and drives the car to safety or some other solution the company can devise for such eventualities. Therefore, the research design and methodology established will allow for data capture and collection for analysis at the post-testing stage of the project. Real-world data on lane correction or any other driver-assist systems are not as well defined as needed for comparing to results captured by a research team, which are usually specific and narrow, and expose issues that are

particular to a vehicle make or model. The National Highway Traffic Safety Administration (NHTSA) said that the first-of-its-kind data specific to traffic accidents that were acquired in 2021 do not yet have proper context and are only meant to be a guide to quickly identify potential defect trends and help determine whether the systems are improving the safety of vehicles [21]. Nonetheless, the fatal error found in this experiment can be reasonably regarded as a potentially dangerous problem for traffic safety.

An experiment conducted in a real-world environment has some advantages and disadvantages. The disadvantage is the inability to control the environment. The advantage is that we were testing the vehicle in its natural environment, and any error that presents itself will be evident to the test driver. A physical anomaly can be detected and corrected. The main concern for this study is the careful introduction of advanced technology where the worldwide average of traffic accidents does not go up when it should decrease. For example, in Japan, 38.9% of single-vehicle crashes involve unintentional lane departure. In the USA, the road departure problem accounts for 41% of all vehicle fatalities [22]. If a lane correction module is not consistent, how would a fatal error that occurs randomly contribute to accidents? The progress made in this study is a significant beginning of a scientific method in experimental test driving in automotive computer technology, yet this is specific to hyperconnected vehicles, whether electric or hydrogen.

Vehicular systems are mostly regulated and controlled by computer functions and applications. Any autonomous driving controller requires accurate sensory information as its input. To meet the requirements, sensors must go through a complex offline calibration process to transfer the raw data in the sensor frame to meaningful data in the vehicle frame [23]. The importance of stress testing vehicles, even if they have come off the assembly line, might not have been adequately recognized for fatal errors. All of the propositions for environmental control and the amount of testing will require testing beyond the constraints of this stress test. Despite this being an early stage in the research, we have set a guideline for further research in a controlled setting where the possibility of interference in real-world traffic is nullified and extraneous variables are reduced significantly. Regardless, the error discovered is significant, even if the issue is in select Tesla models, trims, or years instead of across the entire fleet.

This study did not have access to Tesla's internal data sets, source code, or vehicular performance testing results concerning the lane correction model within the Autopilot Suite. Regardless, this study should provide the vehicle manufacturer with significant feedback on new methods of discovering errors in performance features that can be discovered during the use of the vehicle in a real-world setting. This study's proposed solution for a fix is a software update after a code review. If the issue is persistent, the issue could be a hardware/software problem combination and require a physical replacement of the vision system. The results from this research should start a discussion among governmental regulatory bodies and at the Vehicle Research and Test Center at the NHTSA, where electric vehicles with hyperconnectivity to cyberspace can be evaluated externally to vehicle manufacturers and improve human security for next-generation transportation systems. The findings should be retested in the future on Tesla Models S, X, Y, Roadster, and Semi as it can be reasonably deduced that the lane correction module uses the same proprietary source code, yet this might not be the case.

**Author Contributions:** Conceptualization: J.L. and B.P.R.; methodology: B.P.R.; validation: J.L.; investigation: J.L. and B.P.R.; writing—original draft preparation: J.L. and B.P.R.; writing—review and editing: B.P.R. and E.M.D.; visualization: J.L. and B.P.R.; supervision: B.P.R. All authors have read and agreed to the published version of the manuscript.

**Funding:** This research received no external funding.

**Data Availability Statement:** Not applicable.

**Conflicts of Interest:** The authors declare no conflicts of interest.

## Abbreviations

The following abbreviations are used in this manuscript:

| | |
|---|---|
| AI | Artificial Intelligence |
| ML | Machine Learning |
| FSD | Full Self-Driving |
| ADAS | Advanced Driver-Assistance Systems |
| SUT | System under Test |
| IP | Intellectual Property |
| NHTSA | National Highway Traffic Safety Administration |

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
