# Peer review of "Performance Evaluation of a Lane Correction Module Stress Test: A Field Test of Tesla Model 3"

_futureinternet, doi:10.3390/fi15040138_

Round 1

Reviewer 1 Report

1. The conclusions of the manuscript are rather fatal for Tesla. Therefore, it would be of worth to add: 

- a comparative case for another car model.

- to propose your suppositions, reasons, for the bad results.

2. Please compare your results versus other papers already published in the literature, suitable for the same car model.

3. please describe details of the experimental arrangement, to permit other teams to perform them.

Reviewer 2 Report

Dear Authors,

The paper presented an interesting case study that identified the issues with the lane correction module of Tesla Model 3. 

Following are my comments:

Please provide more details regarding the test. Where was it carried out? Did it include actual traffic?

What is the total sample size of the data? 

Is there any real-world data available for comparison?

What are the proposed solutions to fix the issues with the lane correction module?

In tables 1 and 4, why is the test data classified as 1.0.1, 6.01, 7.01, etc? 

If possible, could you present the results in graphical format? This will make the paper more appealing and easy to understand.

 The paper has too much text. If possible, try to reduce the redundant information.

Round 2

Reviewer 1 Report

None.

Reviewer 2 Report

Dear Authors,

Thank you for answering all my queries.